# Changes in Aggressive Behavior, Cortisol and Brain Monoamines during the Formation of Social Hierarchy in Black Rockfish (*Sebastes schlegelii*)

**DOI:** 10.3390/ani10122357

**Published:** 2020-12-10

**Authors:** Xiuwen Xu, Zonghang Zhang, Haoyu Guo, Jianguang Qin, Xiumei Zhang

**Affiliations:** 1The Key Laboratory of Mariculture, Ministry of Education, Ocean University of China, Qingdao 266003, China; xiuwenxu1207@163.com (X.X.); zonghangzhang@163.com (Z.Z.); 2Fisheries College, Zhejiang Ocean University, Zhoushan 316022, China; haoyuguo@163.com; 3Laboratory for Marine Fisheries Science and Food Production Processes, Qingdao National Laboratory for Marine Science and Technology, Qingdao 266237, China; 4College of Science and Engineering, Flinders University, GPO Box 2100, Adelaide, SA 5001, Australia; jian.qin@flinders.edu.au

**Keywords:** aggressive interactions, aggression, cortisol, monoamines, social hierarchy, black rockfish

## Abstract

**Simple Summary:**

Black rockfish (*Sebastes schlegelii*) are naturally solitary animals. Under a high stocking density in aquaculture, juvenile black rockfish frequently show aggressive interactions, resulting in welfare issues (e.g., body injury). Aggressive interactions can lead to a social hierarchy and influence the responses of animal behavior and physiology. However, the responses of fish behavior and physiology to social hierarchy is limited. This study examined the differences in the growth performance, aggression, cortisol level, brain serotonergic activity, and brain dopamine activity between the dominant individuals and the subordinate individuals in the short-term contest and in the long-term contest. Ultimately, our results suggest that subordinate hierarchy inhibits aggression but does not impact growth in black rockfish. The cortisol-related change in brain monoaminergic activity could be a potential indicator to predict aggressive behavior in black rockfish in captivity with an obvious social hierarchy. This study has provided new insight into the understanding of the regulatory mechanisms between social hierarchy and aggressive behavior in black rockfish, and contributed to building the theoretical basis for behavioral management to solve the welfare issues in captive fish.

**Abstract:**

Aggressive interactions can lead to a social hierarchy and influence the responses of animal behavior and physiology. However, our understanding on the changes of fish behavior and physiology during the process of social hierarchical formation is limited. To explore the responses of fish behavior and physiology to social hierarchy, we examined the differences in the growth performance, aggression, cortisol level, brain serotonergic activity, and brain dopamine activity between the dominant individuals and the subordinate individuals of black rockfish (*Sebastes schlegelii*) in two time scenarios. In the short-term contest, the cortisol level and the ratio of telencephalic 5-hydroxyindoleacetic acid (5-HIAA)/5-hydroxytryptamine (5-HT) was significantly higher in subordinate individuals than in dominant individuals. In the long-term contest, the ratios of 5-HIAA/5-HT in all brain regions were significantly higher, and the frequency of aggressive acts were significantly lower in subordinate individuals than in dominant individuals. In contrast, no difference was detected in growth performance. Significant positive correlations between the cortisol level and serotonergic activity were observed in the short-term contest, but the serotonergic activity was negatively correlated with the aggressive behavior in the long-term contest. These results suggest that subordinate hierarchy inhibits aggression but does not impact growth in black rockfish. The cortisol-related change in brain monoaminergic activity could be a potential indicator to predict aggressive behavior in black rockfish in captivity with an obvious social hierarchy.

## 1. Introduction

Aggressive interactions can lead to a social hierarchy and influence behavior and physiology in vertebrates from teleost to mammals [1,2,3,4,5]. A general behavioral inhibition can cause appetite suppression, reduction of reproductive behavior, and low spontaneous locomotor activities, particularly in subordinate individuals when a social hierarchy exists in an animal population such as arctic charr (*Salvelinus alpinus*) and tree shrews (*Tupaiidea*) [6,7,8,9]. On the other hand, the levels of the monoamine neurotransmitter serotonin (5-hydroxytryptamine, 5-HT) and dopamine (DA) are related to the level of social stress in many species [10,11,12]. In a stable subordinate–dominant relationship, subordinates are socially defeated individuals and generally have a higher level of brain serotonin (5-HT) than the dominant individuals in cichlid fish, *Astatotilapia burtoni,* and pigs *Sus scrofa* [4,13]. However, patterns may differ on the role of brain catecholamine (e.g., DA) from brain serotonin (5-HT) in behavioral response to social stress [4,14]. The brain serotonergic activity is high in socially subordinate *Anolis carolinensis*, whereas the brain dopaminergic activity is low in subordinate fish after an aggressive interaction in lizards, *Sceloporus jarrovi* [15,16,17]. The role of 5-HT in inhibiting aggressive behavior has been well documented in a wide range of vertebrates, including rainbow trout (*Oncorhynchus mykiss*), the lizard, pigs, sparrows (*Passer*), and humans. While the elevated motivation for aggression appears to be the province of dopaminergic action [18]. 

The subordinate fish are characterized by a higher level of plasma cortisol than those of dominant fish in a short-term contest [19,20]. The hypothalamic-pituitary-interregnal (HPI) axis is the major endocrine stress axis of fish, which is homologous to the hypothalamic–pituitary–adrenal (HPA) axis in mammals [21]. Cortisol, the final hormone of this axis, is involved in a suite of physiological maintenance functions including humoral balance, respiration, and energy metabolism. The effects of corticosteroids are mediated through intracellular glucocorticoid (GR) and mineralocorticoid receptors (MR) that act as ligand-dependent transcription factors [22,23]. Teleost fish express both GRs and MRs in the brain except for the hypothalamus, suggesting that cortisol functions in different brain regions [23]. The elevated cortisol level in fish could be an indicator of a poor physiological condition, which may result in a competitive disadvantage [24]. These marked elevations of the corticosteroid stress hormone, cortisol, have been widely used as a social stress index in fish [25,26]. It has been discerned that the differences in cortisol levels between dominant and subordinate individuals were both the consequence and cause of social hierarchy in many fish such as rainbow trout and brown trout [27]. Specifically, the subordinate fish exhibits increasing levels of cortisol after aggressive interactions. In turn, a fish with a high cortisol level is more likely to become the subordinate ones (loser). Thus, the behavioral consequence of social hierarchy is thought to be related to the change in cortisol levels [1,28].

Two possible mechanisms have been proposed in previous studies regarding the relationship between stress level and social hierarchy in fish [18,27]. First, the elevation of cortisol levels can directly influence social behavior. Alternatively, the elevation of cortisol levels can indirectly modulate social behavior by influencing brain monoaminergic activity. However, previous studies have mainly focused on behavior, cortisol, and brain monoamine response to aggressive interactions either for a few hours or for several weeks. It is not clear on the change of cortisol level and brain monoaminergic activity between a short-term to a long-term of aggressive interactions in fish. 

The black rockfish (*Sebastes schlegelii*) is a solitary fish in nature but is widely distributed along the coastal waters of China, the Korean peninsula, and Japan [29]. Under a high stocking density in aquaculture, juvenile black rockfish frequently show aggressive interactions, cannibalism, and so on, resulting in welfare issues (e.g., body injury, mortality, and growth heterogeneity) [30,31,32,33,34]. Thus, to reduce the aggressive behavior of black rockfish at a high density in a captive condition, it is important to understand the impact of aggressive interactions on fish behavior and physiology and explore the physiological mechanism that regulates aggressive behavior. 

In the present study, we investigated the changes in cortisol concentration, aggression, growth performance, and brain monoaminergic activity during the process of establishing a stable social hierarchy in black rockfish. To explore the possible relationship between social hierarchy and aggressive behaviors, we compared the correlation between cortisol concentrations, aggressive behavior, and brain monoaminergic activities between short-term and long-term social interactions in black rockfish.

## 2. Materials and Methods 

All procedures in this study complied with ethical protocols (ARRIVE guidelines; EU Directive 2010/63/EU for animal experiments) approved by the Institutional Animal Care and Use Committee of Ocean University of China.

### 2.1. Fish Materials

Juvenile black rockfish *Sebastes schlegelii* (4 months old) were obtained from the Jinshatan National Fish Hatchery (Qingdao, Shandong, China) and reared in a 5000-L concrete tank. The tank was provided with filtrated seawater at a flow rate of 10 L/min under constant aeration. The temperature was at 18.5–22.7 °C and the photoperiod followed the natural day–night cycle. The light intensity below the water surface was 18–150 Lx. All fish were fed to apparent satiation twice daily with dry pellets (Cheruf, Longxing Feed Corporation, Qingdao, Shandong, China). The main ingredients of the dry pellets are as follows: moisture, ≤10.0%; crude protein, ≥48.0%; crude lipid, ≥9.0%; crude ash, ≤17.0%; crude fiber, ≤2.0%; total phosphorus, 1.5–3.0%; lysine, ≥2.5%.

### 2.2. Experimental Design

In this experiment, dyadic fighting (DF) was used to stimulate aggressive interaction as described by Oliveira et al. (2011) [35]. Twenty-four 36-L glass tanks (long × wide × deep = 40 × 30 × 30 cm) were placed in the laboratory, and each tank was equally divided into two compartments by an opaque PVC separator. Forty-eight black rockfish were selected from the concrete tanks and weighed to form matching pairs (length difference < 2 mm and bodyweight difference < 4%; length ± SD = 4.735 ± 0.496 cm, weight ± SD = 3.067 ± 0.804 g, *n* = 48). After being weighed, the paired fish were placed in each separate compartment in the same glass tank for a 5-day acclimation. On Day 6, the opaque PVC separators were gently removed to allow aggressive interactions to happen. 

For the short-term contest, the fish of twelve glass tanks were allowed to interact for 2 h, a duration that exceeded the necessary time to determine a winner and a loser of the contest. The fighting was featured by frequent biting, chasing, and frightening, resulting in one fish escaping from the fighting arena [31,36]. The winner occupied the central area and the loser tended to hide in the tank edge after fighting. Based on their relative position in the glass tank, the dominant individual (winner) and the subordinate (loser) were readily identifiable in all pairs. After the interaction, each pair of the fish has separated again by placing back the opaque PVC separators for a subsequent aggression test. Mirror image stimulation (MIS) was used to test fish aggression as described by Oliveira et al. (2005) [37]. Briefly, after a 4-h separating period for acclimation, each of the opaque PVC separators was replaced by a double-sided mirror and the fish in the tank could interact with its image for 2 h. All the interactions were videotaped to record aggressive acts for behavioral analyses. After the aggression test, all the fish were tissue sampled for physiological analysis. 

For the long-term contest, the fish in the remaining 12 glass tanks were allowed to interact continuously for four weeks. Each pair in a tank were observed for 5 min daily to check for the dominance–subordination relationship. Individuals were identified based on their skin patterns and swimming tracks, and only the fish with a consistent social hierarchy was used to conduct the aggression test and then sampled for physiological analysis (Figure 1a). 

The experimental condition was similar to that in the concrete tanks. All the fish during the experiment were fed commercial dry pellets by hand twice daily at 0700 h and 1900 h. The pellets were dropped until fish reached apparent satiation and the feeding process lasted for approximately 15 min each time.

### 2.3. Behavioral Observations

All the interactions of black rockfish in the aggression test (MIS) were recorded by video cameras (HDR-AS100 V, SONY Electronics South China Co., LTD, Guangzhou, China) mounted in front of the glass tanks. A 20-min observation of each recording of interactions was randomly selected to identify the aggressive acts. Representative aggressive acts included aggressive displays, demonstration displays, threat displays, and attacks against the mirror image (Figure 1b). Demonstration display: Fish were close to the opponent (0.5 to 1.5 body length) and flaring its body flank to each other with opercula opened, fins erected and body swayed. Threat display: Fish swam quickly to the opponent and then returned to the original place without physical contact [31]. 

### 2.4. Tissue Sampling and Physiological Analysis

The sampling procedure was conducted by the method described in a previous paper [31]. Fish were rapidly netted and anesthetized in 80 mg/L tricaine methanesulfonate (886-86-2, TCI AMERICA, Portland, OR, USA). All fish were individually weighed to the nearest 0.01 g and measured for the fork length to the nearest 0.01 cm. Then, the whole visceral mass was extracted, washed thoroughly with ice-cold 0.86% NaCl solution, and frozen quickly in liquid nitrogen. After collecting the visceral mass samples, the fish were decapitated and their whole brains were quickly dissected into four different brain regions—telencephalon (excluding the olfactory bulb), hypothalamus, optic tectum, and hindbrain—and then stored at −80 °C until the brain monoaminergic activity analysis. All the sampled fish were individually dissected within 45 s. 

The frozen samples were homogenized in 0.86% (wt/vol) NaCl solution with a mechanical homogenizer (scipu001395, IKA Works Guangzhou, Guangzhou, China). The resulting crude extract was centrifuged at 10,000 rpm for 10 min in a refrigerated centrifuge (4 °C), and the supernatant was carefully collected for biochemical analysis. The cortisol, serotonin (5-hydroxytryptamine, 5-HT), 5-hydroxyindoleacetic acid (5-HIAA), dopamine (DA) concentrations were measured using commercial enzyme linked immunosorbent assay (ELISA) Assay Kits (Nanjing Jiancheng Bioengineering Institute, Nanjing, China), and the 3, 4-dihydroxyphenylacetic acid (DOPAC) concentrations were measured using commercial ELISA Assay Kits (Shanghai Yuanye Bioengineering Institute, Shanghai, China) according to the manufacturer’s guidelines. These ELISA Assay Kits have been validated previously [32,38,39,40,41].

### 2.5. Data Statistics

Values presented in the text, tables, and figures were means ± standard error (S.E.). Differences were set to be significant at *p* < 0.05. The data that were normally distributed with equal variance were analyzed by one-way analysis of variance (ANOVA) followed by Duncan’s multiple range post hoc test. Wherever necessary, the data were transformed (square root, exponential, or logarithm) to ensure that the normal distribution and/or homogeneity of variance were obtained. Correlations between monoamine/metabolite ratios and cortisol or aggressive acts were tested by the Spearman rank test. All statistical analyses were performed using SPSS 17.0 for Windows (SPSS Inc., Chicago, IL, USA).

## 3. Results

### 3.1. Growth Performance, Aggression, and Cortisol Levels 

No difference was detected in fish length and weight between the dominant and the subordinate individuals in short-term contest (length *p* = 0.769; weight *p* = 0.890) and long-term contest (length *p* = 0.904; weight *p* = 0.421) (Standard length, one-way ANOVA, F (3, 44) = 28.813, *p* < 0.001; Body mass, one-way ANOVA, F (3, 44) = 55.864, *p* < 0.001, Table 1). The number of aggressive acts was not different between dominant (winner) and subordinate (loser) individuals in the short-term contest (*p* = 0.777) but was significantly higher in dominant individuals than in subordinate individuals in the long-term contest (*p* < 0.05) (one-way ANOVA, F (3, 44) = 3.26, *p* < 0.05, Table 1). The cortisol level was significantly higher in subordinate individuals than in dominant individuals in the short-term contest (*p* < 0.001), whereas no difference was found between subordinate individuals and dominant individuals in the long-term contest (*p* = 0.928) (one-way ANOVA, F (3, 44) = 313.68, *p* < 0.001, Table 1). 

### 3.2. Brain Monoaminergic Activity

We used the Spearman rank test to analyze the relationships between plasma cortisol and brain monoaminergic activity in short-term contests and the relationships between the mean number of aggressive acts and brain monoaminergic activity in long-term contest. In the short-term contest, the ratio of 5-HIAA/5-HT in the telencephalon was significantly higher in subordinate individuals than in dominant individuals (*p* < 0.05). In the long-term contest, the ratio of 5-HIAA/5-HT in all brain regions was significantly higher in subordinate individuals than in dominant individuals (*p* < 0.05; Figure 2). 

The ratio of DOPAC/DA in the hypothalamus was significantly higher in dominant individuals than in subordinate individuals in the short-term contest (*p* < 0.05), whereas it is similar in all brain regions between subordinate individuals and dominant individuals in the long-term contest (*p* > 0.05; Figure 3).

### 3.3. Correlations between Monoamine/Metabolite Ratios and Cortisol or Aggressive Acts

As the cortisol levels were different between subordinate individuals and dominant individuals only in the short-term contest, while the number of aggressive acts was different only in long-term contest. We tested the relationships between plasma cortisol and brain monoaminergic activity in short-term contests and relationships between the mean number of aggressive acts and brain monoaminergic activity in long-term contest. Specifically, in short-term contests, cortisol levels had a significant positive correlation with the telencephalon 5-HIAA/5-HT ratios in subordinate individuals (*p* < 0.001; Figure 4). 

In a long-term contest, the number of aggressive acts had a significantly negative correlation with the hindbrain 5-HIAA/5-HT ratios in subordinate individuals (*p* < 0.05; Figure 5).

## 4. Discussion

No difference in growth performance has been found between the dominant and subordinate individuals of black rockfish after a long-term contest, suggesting that social hierarchy may not influence the growth of black rockfish. This result is consistent with previous studies when the juvenile black rockfish were fed to apparent satiation throughout the experiment [30]. In contrast, McCarthy observed that subordinate rainbow trout are generally excluded from preferential access to food, resulting in a reduction of fish growth at a food ration of 0.5 B.W. day^−1^ [42]. However, this may be due to the reason that fish did not receive enough food during the trial. Besides, although the dominant individuals had more advantages to compete for feed, the energy consumed might be expended for aggression. Subordinate individuals may take less feed, but the low frequency of attacking behavior may spare energy for growth [29]. Therefore, there was no difference in growth performance between dominants and subordinates in the 1-month short-term experiment. Our results show that whether social hierarchy affects fish growth may be related to the availability of adequate food.

Competitive ability is a key factor to determine the outcome of a competition in fish such as brown trout and juvenile steelhead trout [43,44]. Although the innate aggressive tendency is an indicator of competitive ability, other factors including body size, physiological condition, and prior fighting experience also play a vital role to determine fish performance during the period of establishing a stable social hierarchy [45,46,47]. So, it is not surprising to find no difference in aggression between the winner and loser in the short-term contest in the present study. However, a decrease in aggressive acts during mirror interaction was observed in subordinate individuals after a long-term contest. It indicates that the subordinate hierarchy may inhibit aggression in black rockfish. The results of the present study support a view that the establishment of social hierarchy is an evolutionary adaptation that can prevent the need for continued aggression and associated risk of injury and that have fitness effects on the lifelong success of individuals [48,49,50]. Our study shows that it may be the result of social repression that the subordinate individuals are less aggressive than the dominant individuals.

In addition, two possible pathways have been envisaged for the mechanism through which social hierarchy influences aggression. Elevated cortisol levels could influence social behavior directly. Alternatively, elevated cortisol levels could modulate social behavior indirectly by influencing brain monoaminergic activity [19,27]. The differences in cortisol levels between dominant and subordinate individuals were both the cause and consequence of social hierarchy in many fish such as rainbow trout and brown trout [27]. Specifically, fish with a high cortisol level are more likely to become the subordinate ones (loser). In turn, the subordinate fish exhibits increasing levels of cortisol after a short-term contest, which is in line with the result of the present study. However, the social hierarchy established after the long-term contest did not affect the cortisol response in black rockfish. The results of the long term trial in this study do not seem to support the hypothesis that social hierarchy inhibits aggression and the elevated cortisol could directly impact competitive ability. 

The metabolites 5-HIAA and DOPAC are formed following the re-uptake of the parent monoamines (5-HT and DA, respectively) from the synaptic cleft, and their accumulation in neural tissue is probably time-dependent. Thus, the analysis of tissue concentrations of neurotransmitter metabolites does not reflect the instantaneous neural activity. Therefore, the brain metabolite/monoamine ratio was considered an indicator of increased monoamine utilization [36]. In the present study, the 5-HIAA/5-HT ratio in the telencephalon was increased in subordinate individuals after the short-term contest compared to dominant individuals, while the 5-HIAA/5-HT ratio was increased in all brain regions after the long-term contest. Thus, it appears that the brain serotonin system was activated during aggressive interactions, leading to a gradual increase in metabolite levels over time. This result is similar to the report of the influence of aggressive interactions on brain monoaminergic activity in reptiles such as lizards [16]. In either fish or reptile, subordinate individuals can exhibit a substantial activation of the 5-HT system and chronically increase serotonergic activity during aggressive encounters. 

In the present study, significant correlations between the cortisol level and 5-HIAA/5-HT ratios were observed after the short-term contest. The response of brain monoamine to social stress has been documented in mammals such as pigs, but rarely in fish [4,36]. A significant increase in serotonergic activity (ratio of 5-HIAA/5-HT) was detected in the telencephalon with cortisol treatment in rainbow trout [46]. It seems that the cortisol level can regulate brain monoaminergic activity under social stress. In addition, the serotonin level inhibits aggression in mammals such as rats (*Rattus norvegicus*) and reptiles such as lizards [18,51]. As expected, the serotonergic activity (ratio of 5-HIAA/5-HT) was negatively correlated with aggressive acts in the present study, and the more subordinate the acts exhibited by a fish, the higher its 5-HIAA/5-HT ratio. The present study suggests that the cortisol-related change in brain monoaminergic activity might be a potential regulatory pathway for the social hierarchy to influence aggression in black rockfish. However, more studies are needed to establish the link between the HPI axis function and brain noradrenergic activity over time. 

The ratio of DOPAC/DA in dominant individuals was higher than in subordinate individuals at the end of the short-term contest, but this result did not occur at the end of the long-term contest. Our study suggests that dopaminergic activation occurs in the dominant fish when the fighting is initiated during the early stage of social hierarchy formation. Although the dopamine system is associated with the formation of social hierarchy in other fish such as arctic charr [51,52], this does not apply to black rockfish. 

## 5. Conclusions

Under satiated feeding, the subordinate hierarchy inhibits aggression but does not impact the growth performance in black rockfish, which supports the view that social hierarchy is an adaptation that conveys evolutionary fitness. Our results reveal that the cortisol-related change in brain monoaminergic activity could be a potential indicator to predict aggressive behavior in black rockfish in captivity with an obvious social hierarchy. This study has provided new insight into the understanding of the regulatory mechanisms between social hierarchy and aggressive behavior in black rockfish, and contributed to building the theoretical basis for behavioral management to solve the welfare issues in captive fish. Artificially adding the appropriate number and size of relatively large fish to cause the social repression of the farmed fish to control their aggression is an area worth exploring.

## Figures and Tables

**Figure 1 animals-10-02357-f001:**
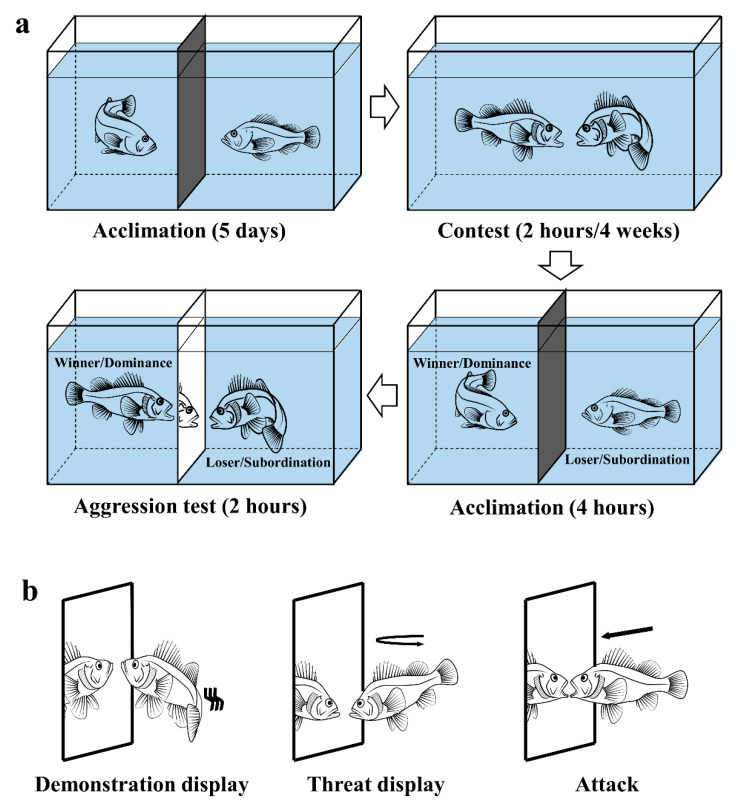
Schematic drawing of (**a**) the experimental tanks and (**b**) the representative aggressive acts of black rockfish in mirror interaction (aggression test).

**Figure 2 animals-10-02357-f002:**
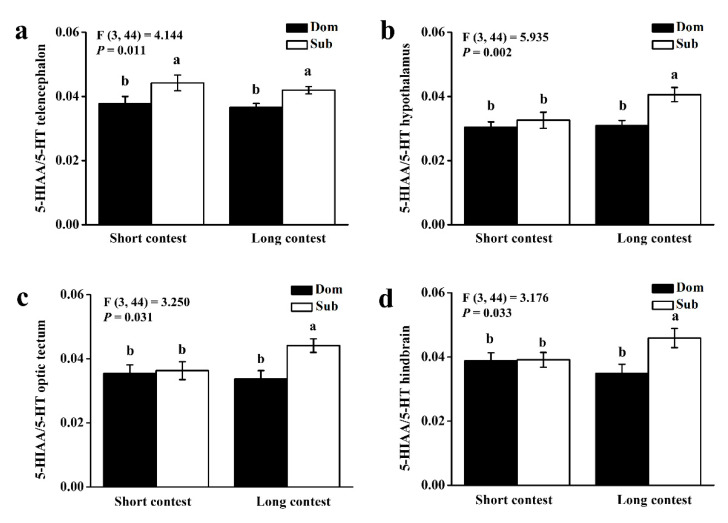
5-hydroxyindoleacetic acid (5-HIAA)/5-hydroxytryptamine (5-HT) ratios of (**a**) telencephalon, (**b**) hypothalamus, (**c**) optic tectum, and (**d**) hindbrain in dominant (Dom) and subordinate (Sub) black rockfish from short-term contest to long-term contest. Different letters indicate significant differences among these four groups, which were detected by one-way ANOVA (*p* < 0.05). Values are mean ± S.E. (*n* = 12).

**Figure 3 animals-10-02357-f003:**
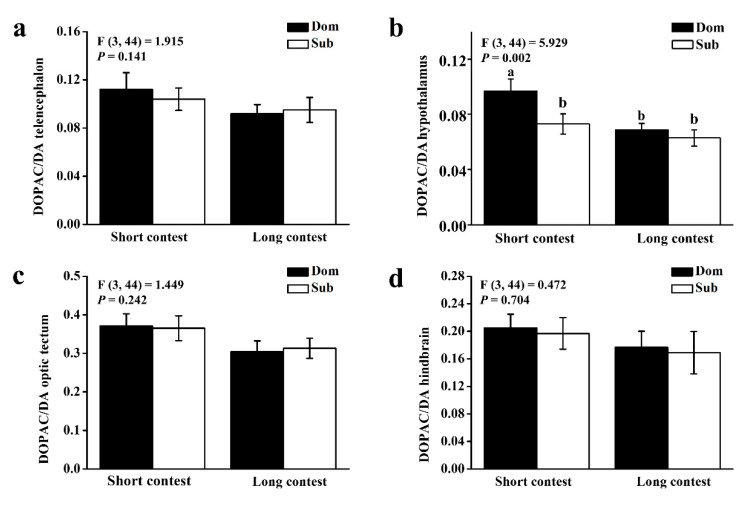
3, 4-dihydroxyphenylacetic acid (DOPAC)/dopamine (DA) ratios of (**a**) telencephalon, (**b**) hypothalamus, (**c**) optic tectum, and (**d**) hindbrain in dominant (Dom) and subordinate (Sub) black rockfish from short-term contest to long-term contest. Different letters indicate significant differences among these four groups, which were detected by one-way ANOVA (*p* < 0.05). Values are mean ± S.E. (*n* = 12).

**Figure 4 animals-10-02357-f004:**
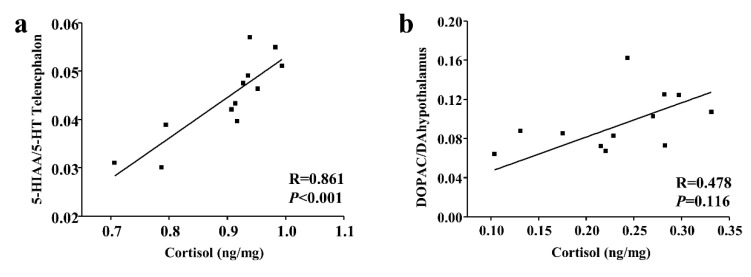
Relationship between cortisol and (**a**) 5-HIAA/5-HT ratios of subordinate black rockfish in telencephalon and (**b**) DOPAC/DA ratios of dominant black rockfish in hypothalamus after a short-term contest. Spearman R and *p* values are given.

**Figure 5 animals-10-02357-f005:**
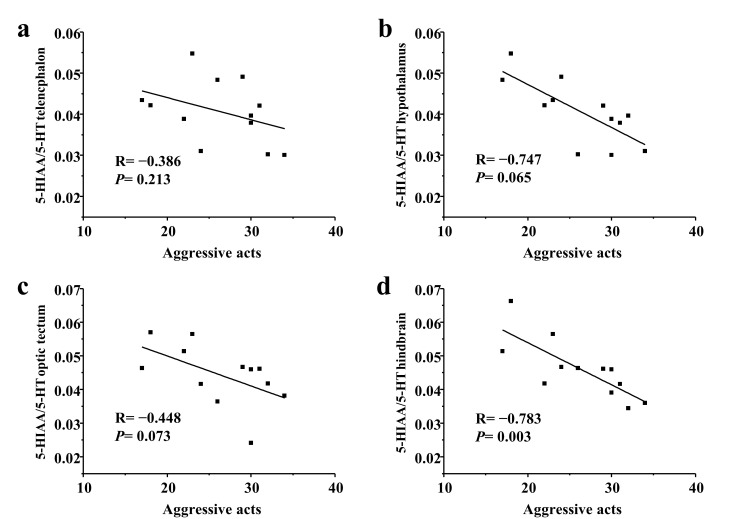
Relationship between aggressive acts and 5-HIAA/5-HT ratios of subordinate black rockfish in (**a**) telencephalon, (**b**) hypothalamus, (**c**) optic tectum, and (**d**) hindbrain after a long-term contest. Spearman R and *p* values are given.

**Table 1 animals-10-02357-t001:** Cortisol levels, aggression (aggressive acts per 20 min) and growth performance in dominant (Dom) and subordinate (Sub) black rockfish. Different letters indicate significant difference among these four groups, which were detected by one-way analysis of variance (ANOVA) (*p* < 0.05). Values are mean ± S.E. (*n* = 12).

Aggressive Interactions	Fish	Standard Length (cm)	Body Mass (g)	Aggression (acts/20 min)	Cortisol (ng/mg)
Short-term contest	Dom	4.57 ± 0.12 ^b^	2.83 ± 0.21 ^b^	39.6 ± 2.16 ^a^	0.24 ± 0.012 ^b^
	Sub	4.63 ± 0.14 ^b^	2.81 ± 0.22 ^b^	40.3 ± 4.33 ^a^	0.92 ± 0.023 ^a^
Long-term contest	Dom	6.33 ± 0.15 ^a^	6.73 ± 0.35 ^a^	42.0 ± 3.63 ^a^	0.23 ± 0.009 ^b^
	Sub	6.22 ± 0.28 ^a^	6.37 ± 0.31 ^a^	28.6 ± 2.81 ^b^	0.23 ± 0.016 ^b^
*p* value		< 0.001	< 0.001	< 0.05	< 0.001

Superscript a, b: Different letters indicate significant difference among different groups within the same column.

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
