# Peer review of "Changes in Aggressive Behavior, Cortisol and Brain Monoamines during the Formation of Social Hierarchy in Black Rockfish (Sebastes schlegelii)"

_animals, 2020, doi:10.3390/ani10122357_

Round 1

Reviewer 1 Report

The Manuscript entitled "Changes in aggressive behavior, cortisol and brain monoamines during the formation of social hierarchy in black rockfish (Sebastes schlegelii)" is really complete study about the behavior in fish which can be an important source of information about social interactions. In addition, some interesting experiments are presented to evaluate the results globally. But at the same time, the fact that the assay does not specify more HPI axis markers sometimes leads to some difficulties in understanding.

Introduction: I think you should introduce the HPI axis if you are going to study the monoaminergic and cortisol system.

Material and methods:

Line 139: What tricaine methanesulfonate have they used?

Line 149-152: Indicate that the Elisa kit has been used for all studies (cortisol, serotonin…) and refers to paper (28). But, in the paper 28, the cortisol analysis is through Iodine radioimmunoassay kits from other company. The protocol used should be clearer.

Results:

Table 1: It is not clear what is comparing: short-term contest vs long-term contest or dom vs sub?

Discussion:

Line 217-220: So, did McCarthy´s experiments last much longer? You should specify the percentage of food that was supplied.

Line 267-269: ¨The present study suggests that the cortisol-induced change in brain monoaminergic activity might be a potential regulatory pathway for the social hierarchy to influence aggression in black rockfish. However, more studies are needed to establish the link between the HPI axis function and brain noradrenergic activity over time¨

Why do they suggest that cortisol causes changes in the monoaminergic activity and not the other way around? On the HPI axis in fish, various neurotransmitters (serotonin, dopamine or norepinephrine) are known to be involved, in addition to neurohormones. Which stimulate the secretion of ACTH, and ACTH stimulates the release of cortisol into the bloodstream, in stressful situations. For this reason I do not understand how with your data, you can give that conclusion. And I would like you to explain it to me better, thank you.

Line 271-274: Can you explain these results to me better? I don't understand why there are no significant changes; and there are only short-term contest changes in the hypothalamus (no in telencephalon, optic tectum or hindbrain). What do you think this difference is due to?

Author Response

Dear Reviewer,

Reviewer 2 Report

Manuscript ID: animals-982558

Title: Changes in aggressive behavior, cortisol and brain monoamines during the formation of social hierarchy in black rockfish (Sebastes schlegelii)

Authors: Xiao Yan, Chao-Bin Qin, Guo-Kun Yang, Da-Peng Deng, Li-Ping Yang, Jun-Chang Feng, Jia-Li Mi, Guo-Xing Nie * Submitted to section: Aquatic Animals,

This is my review of Manuscript ID: animals-982558. I recommend accept the following the article after correcting all my comments.

Summary and Abstract

Lines

It is not clear to me why a summary is needed as well the Abstract and the different between them. Abstract in my opinion enough.14-21. The summary should be improved so that it is clear. Purposes of the research, methods, results, and meanings.26-39. Please in two sentences it will be understood the research methods for who is reading the abstract.

1.  Introduction

50 - variety of species? in all the vertebrates? Explain in more detail.

61-63- have been widely used as a social stress index. In all the vertebrates or only in fishes? Please write more accurately throughout the article and do not use general words.

70. Please add what is known about cortisol in brain and its involvement in the biochemical process effect on this organs and mechanism its action and knowledge of its receptors in the brain. I think it will improve the knowledge in the field.

77. There are many studies that show in fish the effect of density on growth. Density has different effects here I understand only a few examples that are important to address. 207.  Degani, G. & Levanon, D. (1983). The influence of low density on food adaptation, cannibalism and growth of eels (Anguilla anguilla (L.)). Israel Journal of Aquaculture, 35:53 60 193.          Degani, G., Levanon, D. & Dosoretz, C. (1985). Growth of Anguilla anguilla in different densities in outdoor containers with Tilapia aurea. Prog. Fish. Cult., 47:114 118.

Materials and Methods

94. Add the composition of dry pellets (% protein, % fat and % carbohydrate)

Results

164-171.  Table 1. Please indicate which test was done on each statistical comparison e.g. (P < 0.05, Table 1). Do it in all the results.

194-200. Please add the test use for Correlations between monoamine/metabolite ratios and cortisol or aggressive acts (P and R?).

Discussion

229-224 Add references

214-275. Please do not use in general words, fish but indicate what species of fish the findings are cross-referencing. Pleas in all the discussion. The discussion needs to be improved. Should be clear the findings of this study relate to prior knowledge in the field. What has improved this research is the knowledge. What is the critique of the study. And it is also desirable to note future suggested to improve in the field.

Author Response

Dear Reviewer,

Reviewer 3 Report

Dear authors, 

the manuscript entitled "Changes in aggressive behavior, cortisol and brain 2 monoamines during the formation of social hierarchy 3 in black rockfish (Sebastes schlegelii)". In stocking density in aquaculture, juvenile black rockfish frequently show aggressive interactions, resulting in welfare issues (e.g., body injury). The study examined the changes in the growth performance, aggression, cortisol level, brain serotonergic activity, and brain dopamine activity during the process of social hierarchical formation (short-term aggressive interactions to long-term aggressive interactions).  Results suggest that subordinate hierarchy inhibits aggression but does not impact growth in black rockfish. The cortisol-induced change in brain monoaminergic activity could be a potential indicator to predict aggressive behavior in black rockfish in captivity with an obvious social hierarchy. This study has provided new insight  into the understanding of the regulatory mechanisms between social hierarchy and aggressive behavior in black rockfish, and contributed to building the theoretical basis for behavioral management to solve the welfare issues in captive fish. 

The design is well done; results and discussion are pertinent with the topic. 

English style and language are fine. 

This referee agrees with the conclusions proposed. 

As  a whole it is a well-written manuscript and I accept it in its present form. 

Author Response

Dear Reviewer,

Round 2

Reviewer 1 Report

Many thanks to the authors for answering all the questions asked. I think the work has improved a lot by entering all the information of the HPI axis; now the understanding of the article is much better.

Most of the questions asked, I could understand how the investigation had been carried out, but I considered that it was not written in an adequate way due to lack of information. Because, as the authors have commented, there are different ways of action.In addition, there were serious errors in the material and methods and in the figure legends, which the authors have perfectly modified.

Now I think the article is suitable for publication, thank you.